

# Opening the black box of bird-window collisions: passive video recordings in a residential backyard

Brendon Samuels[1], Brock Fenton[1], Esteban Fernández-Juricic[2] and Scott A. MacDougall-Shackleton[3]

[1] Department of Biology, University of Western Ontario, London, Ontario, Canada
[2] Department of Biological Sciences, Purdue University, West Lafayette, Indiana, United States
[3] Department of Psychology, University of Western Ontario, London, Ontario, Canada

## ABSTRACT

Collisions with windows on buildings are a major source of bird mortality. The current understanding of daytime collisions is limited by a lack of empirical data on how collisions occur in the real world because most data are collected by recording evidence of mortality rather than pre-collision behaviour. Based on published literature suggesting a causal relationship between bird collision risk and the appearance of reflections on glass, the fact that reflections vary in appearance depending on viewing angle, and general principles of object collision kinematics, we hypothesized that the risk and lethality of window collisions may be related to the angle and velocity of birds' flight. We deployed a home security camera system to passively record interactions between common North American bird species and residential windows in a backyard setting over spring, summer and fall seasons over 2 years. We captured 38 events including 29 collisions and nine near-misses in which birds approached the glass but avoided impact. Only two of the collisions resulted in immediate fatality, while 23 birds flew away immediately following impact. Birds approached the glass at variable flight speeds and from a wide range of angles, suggesting that the dynamic appearance of reflections on glass at different times of day may play a causal role in collision risk. Birds that approached the window at higher velocity were more likely to be immediately killed or stunned. Most collisions were not detected by the building occupants and, given that most birds flew away immediately, carcass surveys would only document a small fraction of window collisions. We discuss the implications of characterizing pre-collision behaviour for designing effective collision prevention methods.

## INTRODUCTION

Collisions with plate glass on buildings, transportation shelters, noise barriers and fences are a major source of bird mortality, killing hundreds of millions of birds each year in North America and resulting in uncounted mortality worldwide (*Loss et al., 2014*). Most bird-window collisions in North America occur at residential and low-rise buildings, the

Corresponding author
Brendon Samuels, bsamuel2@uwo.ca

most numerous types of structure (*Machtans, Wedeles & Bayne, 2013*; *Loss et al., 2014*). Developing strategies for managing the risk of bird-window collisions is an emerging priority for bird conservation management. There is mounting public and academic interest in methods for treating plate glass windows with materials that can alert birds to the presence of an obstacle to their passage so that they may avoid flying into it (*Klem, 2021*). However, there are few empirical data to show how birds behave in the moments leading up to a collision or how successful avoidance occurs. Characterizing pre-collision behaviour could have implications for designing collision prevention technology, because the effectiveness of deterrents will depend on birds' abilities to detect an obstacle.

Much of our knowledge about bird-window collisions has been inferred by studying bird mortality using an observational, *after-the-fact* approach. Numerous field studies have involved monitoring or surveying buildings and documenting bird carcasses and injured birds at buildings, including combinations of residential homes, institutional, and mid-rise and high-rise buildings (*e.g.*, see *Machtans, Wedeles & Bayne, 2013*; *Loss et al., 2014* for syntheses). Monitoring studies produce empirical data that can inform conservation management in several respects, such as characterizing direct and indirect anthropogenic causes of mortality and population-level impacts (*e.g.*, *Calvert et al., 2013*; *Loss, Will & Marra, 2015*). Collision monitoring provides data on the particular species that suffer collisions in a given study area (*e.g.*, *Sabo et al., 2016*) and can help to identify locations of specific windows that are prone to collisions. Then, window retrofits can be justified using evidence and targeted around high-risk windows to maximize efficiency. Monitoring studies have helped to characterize the spatial, temporal, and structural factors associated with elevated collision risk for particular regions, buildings, or even specific façades (*e.g.*, *Hager et al., 2017*; *Riding, O'Connell & Loss, 2019*). These studies have also revealed differences in collision risk across bird species. For instance, *Nichols et al. (2018)* analyzed multiple citizen science datasets of collision monitoring to model collision susceptibility of different groups of birds, classifying some as "supercolliders" and others as "superavoiders". They found that nocturnal migrants were more susceptible to collisions than diurnal migrants, and that collision susceptibility for most species was strongly predicted by local abundance. Their top model suggested a connection between collision susceptibility and taxonomy, with genus and species categories predicting collision susceptibility in around 20% of the species represented in their dataset. *Riding, O'Connell & Loss (2019)* studied façades at 16 mid-rise buildings and identified façade-level features that were associated with elevated collision risk, including proportional glass coverage, façade length and façade height, and the overall shape of the façade. Relatively fewer studies have used building surveys to document collision mortality at residential homes, perhaps because residential buildings are subject to privacy considerations (*e.g.*, *Klem, 1990*). In recent years, citizen/community science tools are increasingly used to collect data from observations of bird-window collisions at residential homes reported by the public (*e.g.*, *Bayne, Scobie & Rawson-Clark, 2012*; *Kummer, Bayne & Machtans, 2016*; *FLAP Canada, 2021*).

Observational building surveys provide strong naturalistic validity by documenting collisions in the real world, but they reveal little about how collision events actually

happen. For instance, if a bird is found dead below a window, nothing about the bird's condition can indicate the speed or angle at which it impacted the glass, nor can it describe what the bird was doing before it hit the window, nor the circumstances in which it failed to detect the obstacle. We also cannot generalize from the number of carcasses observed to the number of actual collisions that occurred. It is reasonable to assume that some collisions do not immediately kill or even ground the bird and would therefore not leave behind evidence to suggest that a collision occurred. No research published to date has presented data to indicate the lethality of window collisions—the ratio between the total number of birds that collide with windows and the subset that experience severe injury or death as a result. Thus, we do not know the proportion of birds that strike windows that are ultimately not detected by standardized building survey protocols. Studies have examined differences among glass types and window markings in terms of effectiveness for preventing bird-window collisions by erecting panes of glass at a field site and counting the number of bird carcasses left by collisions (*e.g.*, *Klem & Saenger, 2013*). However, like building surveys, these studies have not captured information about pre-collision behaviour or relative mortality.

Previous research has linked the risk of lethal bird window collisions with environmental variables such as building characteristics, surrounding urbanization and time of day (*e.g.*, *Hager et al., 2017*; *Riding, O'Connell & Loss, 2021*). Yet, it is unclear precisely how the surrounding environment and qualities of windows themselves interact with birds' sensory perception and behaviour to affect collision risk. Reflections on glass vary in appearance and intensity depending on ambient lighting under different weather conditions and at different times of day, as well as illumination behind the glass. The appearance of reflections also varies depending on viewing angle. There are also unexplored potential risk factors related to the physical parameters of bird flight before and during a collision, such as the velocity and angle of approach, as well as the body mass and momentum (the product of mass and velocity) of the bird. The effects of these factors during a collision on the body of a bird may be predicted based on the laws of momentum and modelling of object collisions in general. Bird-window collisions are inelastic, meaning some kinetic energy is always converted upon impact into other forms such as heat or sound. Momentum is conserved in inelastic collisions. When a small, soft object like a bird collides with a larger, stationary object like a window, the total momentum of the two objects before and immediately after impact are equal. During a collision, the objects experience a force over a short time that results in a change in their respective momentum. For collisions that occur in two- or three-dimensional space, the force applied may be influenced by several parameters including the objects' momentum (*i.e.*, mass and velocity) and the angle of the objects' initial velocity vector leading up to impact. A moving, soft object will experience more force in a perpendicular collision as all of the force is directed outward at that object, compared to a glancing collision where only a portion of the force is directed back toward the object. While there is little empirical information to indicate how much of the force generated by a collision is absorbed by the body of a bird, there is reason to expect that angle of a bird's approach to the window as well as its flight velocity will

influence the magnitude of force experienced upon impact and the resulting effects on the bird's tissues.

To gain insight into how bird-window collisions occur at real buildings, we used passive video monitoring to record bird behaviour at windows. Because the behaviour of wild birds and the timing of potential collisions are unpredictable, observing collisions by passive video recording system requires a mechanism for detecting the onset of target events and filtering out noise (*e.g.*, motion or audio detection). We used a commercial home security system in a residential backyard setting to passively record interactions between birds and window glass over 2 years. We analyzed footage to examine birds' behaviour in the moments leading up to potential collisions as well as outcomes of collision events.

Our analyses aimed to test the hypotheses that collision risk and the risk of being killed or incapacitated (*i.e.*, stunned or injured) by a collision with a window, are related to the bird's flight velocity and angle of approach to the window (see above). We predicted that higher flight speed, and angles of approach approximately perpendicular to the window, are more likely to result in injurious or lethal collisions. We also considered time of the day as a potential confounding factor as it can change the reflection of ambient light on the window.

## MATERIALS AND METHODS

### Study area

We filmed in the backyard of a residential property in London, Ontario, Canada from May to October in 2019 and in 2020. The property is located immediately adjacent to a mature eastern deciduous forest nested within a suburban neighbourhood. We expected there to be a relatively high abundance of resident and migratory bird species in study area based on the adjacent forest's designation as Significant Wildlife Habitat by the Province of Ontario, results of long-term window building monitoring conducted within a few kilometers of the site, and the property owners' observations over several years of feeding birds in the backyard. The property owners maintain a variety of bird feeders and bathing stations in the backyard that were included in the setup for the present study. We are aware that at a spatial level, we only gathered information from a single location, so our results and conclusions cannot be generalized to all window collisions contexts. However, we focused this study on the role of variables related to the behaviour of the animals (velocity, angle of approach, *etc.*) that are not well-understood. Therefore, we consider that our study can provide some novel insights for future testing despite the limited spatial representation.

The configuration of the backyard study area is shown in Fig. 1. The backyard contained several bird feeders that were maintained with a combination of seed mixes by the homeowner throughout the study period. For the first year of monitoring, feeders were placed at two positions. In the second year, a third feeder was added by the homeowner positioned closer to one of the windows. The backyard also contained two bird baths as well as a large patio table with an umbrella and chairs. The position of the house relative to the mature trees on the property meant that the south-facing windows under study were

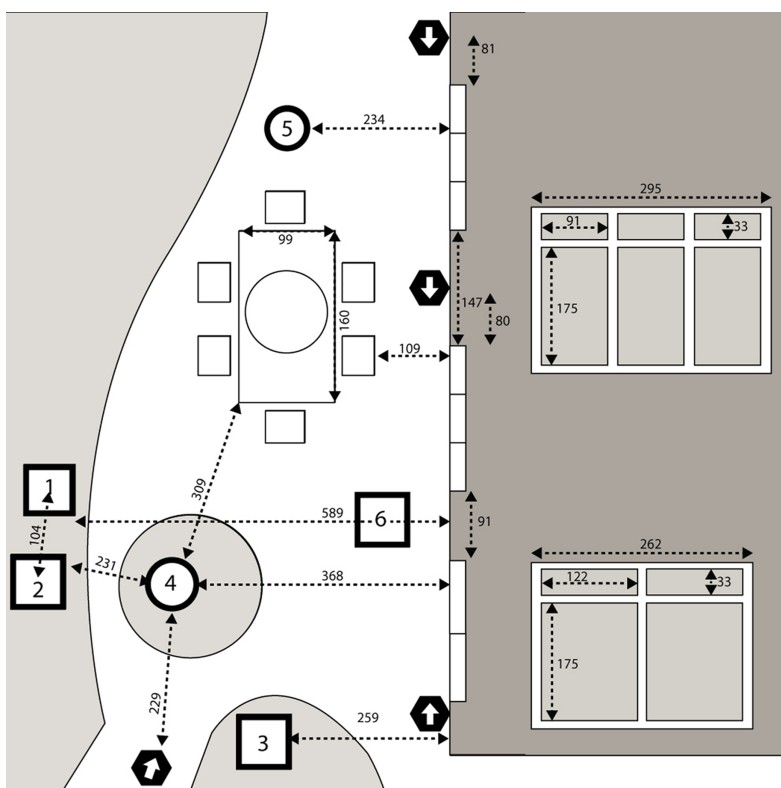

**Figure 1 Overhead view of the backyard study area.** Overhead view of the backyard study area. Drawing is not perfectly to scale. Measurements between points of reference are shown in centimeters. Dark shaded area represents the house with a horizontal view of the transom windows and sliding doors inset on the right with dimensions; the middle and top and middle sets of windows and doors were identical, while the lowest set were of a different size. 1, 2 represent locations of bird feeders stocked with bird seed; 3 represents a hummingbird feeder; 4, 5 represent bird baths; 6 represents an additional feeder added by the homeowner in 2020. Locations of four cameras are indicated by black hexagons with arrows depicting the orientation of the lens.

shaded through most of the day and exposed to direct sunlight only in early morning (approximately 7:00–10:00 am) and late afternoon (approximately 4:00–5:00 pm).

## Camera system

We installed four Arlo Pro 2 wireless home security cameras in the study area (VMS4430P; Netgear, San Jose, CA, USA). The cameras recorded in 1,080 p resolution at 24 frames per second. Fluctuations in wireless signal strength occasionally reduced the resolution to 720 p. We positioned three of the cameras along the back wall of the house immediately adjacent to three sets of large glass sliding doors (Fig. 1). We used brackets to mount these cameras to the wall approximately 2 m above ground adjacent to the windows, with the camera lenses aimed approximately parallel to the nearest window and the ground, and offset from the window frame by approximately 20 cm. We positioned a fourth camera on a tripod (approx. 1.2 m above the ground) near the bird feeders to provide a view of the entire study area. Thus, the configuration of the four cameras with overlapping fields of view allowed for most events to be captured from multiple angles.

The cameras were triggered synchronously to record video of a fixed duration (10–15 s post-trigger, plus 3 s pre-trigger) using a combination of infrared motion detection (*e.g.*, a bird passing in front of the lens) and audio detection (*i.e.*, a spike in amplitude associated with the thud of a bird striking the glass). The sensitivity of each type of trigger was adjusted within each camera to optimize the signal-to-noise ratio and reduce the frequency of recordings being triggered by unrelated environmental events such as falling leaves, background noise or sudden changes in ambient light. Audio detection sensitivity was tested by simulating the sound of a collision by clapping hands in front of the windows. The audio trigger sensitivity of three cameras was set to three or medium; the camera placed on the tripod had audio triggers disabled to avoid detecting sounds produced by birds at the feeders. Motion sensitivity on all four cameras was set to "7" or medium-high. The cameras uploaded footage directly to a cloud server. In addition to the motion and audio-triggered recordings, we programmed two of the cameras to record video on a continuous rolling basis to provide a failsafe in case events failed to trigger the cameras.

## Video analysis

All the video clips that were captured using audio or motion triggers were stored on the cloud server for a period of 7 days. The continuous recorded footage recorded by two of the cameras was stored for 14 days. Footage was inspected independently throughout the study period by the first author and an assistant. Video clips of birds approaching or interacting with the windows were downloaded for permanent storage and included in later analyses.

A single rater analyzed the video clips using a standardized protocol in VLC Media Player (VideoLAN, Paris, France). All video clips that were saved throughout the study period were processed together in one batch. For each saved video clip, events were classified as either a collision or near-miss. An event was coded as a collision if the bird made physical contact with the glass. We excluded any events in which birds contacted the glass repeatedly at close range, which might indicate they were attacking their reflections on the glass rather than colliding. An event was coded as a near miss if the bird initially appeared to fly along a path that would lead to it colliding with a window, but suddenly changed its trajectory (reducing velocity or sharply shifting direction) within 2 m of approaching the building, and subsequently did not make impact. A change in trajectory preceding a near miss was determined by drawing a straight line along the initial flight path of the bird and confirming that the line projected toward a window, then comparing the bird's observed displacement across subsequent frames. Examples of collision and near miss events are provided in the Supplemental Material. We included video frames in the flight analyses that began either when the bird first entered the frame from off-camera and passed a reference point within the measurements of the backyard, or from the last spot where the bird took off from perching within the backyard (*i.e.*, the "starting" position). The frames included in the flight analyses ended either when the bird impacted a window or at the frame when the bird was closest to the window. We estimated the duration of each flight by multiplying the total number of video frames included in the analysis by 0.042 s (the duration between frames at the 24 fps recording setting used by the cameras). We estimated the distance covered by the flight by marking the change in the position of

the bird across consecutive video frames and comparing the difference relative to known dimensions of the study area. For each event that was included in the analyses, video clips recorded by multiple cameras (typically 2 or 3) was examined by the rater in combination to reduce bias in visual approximation of changes in the birds' position over time. The distance traversed by a bird between consecutive video frames was noted in a spreadsheet (Microsoft Excel; Microsoft, Redmond, WA, USA). We then calculated average velocity within each flight as the total distance covered in flight divided by the total flight duration. We compared changes in the position of the bird between video frames with dimensions of the backyard to produce an overhead map showing the approximate flight paths followed by birds as they approached the windows (see Supplemental Material). We visually estimated the horizontal angle of approach (*i.e.*, the azimuth) for each flight path relative to the normal angle from the window (*e.g.*, a head-on approach at exactly 90° from the window would be represented as 0°) to the nearest 5° angle.

To include horizontal angle of approach in our statistical analyses, we used the absolute value of the estimated angle from perpendicular thus controlling for direction (*i.e.*, using values that ignore whether the bird approached from the left or right side of the window).

## Statistical analyses

We ran generalized linear mixed models with the afex R package (*Singmann et al., 2022*) with two dependent variables (probability of collision, probability of visible injury) with the same independent factors: velocity, horizontal angle of approach, and time of the day. We included time of day as an index of variability in ambient lighting conditions that we expected to produce a consistent effect on the appearance of the window glass; we did not directly measure ambient light within the study area. Because we had more than one collision event per species in some data entries and we did not have individuals tagged, we included species as a random factor to account for the possibility of non-independence of samples in the model (*i.e.*, the same individual birds approaching the windows multiple times over the duration of the study). The probability of collision when birds were on a collision approach included two values: 0, no collision (near miss), and 1, collision. The probability of visible injury considered only birds that collided with the window, including two values: 0, animals flew away immediately, and 1, animals were stunned (*i.e.*, landed on the ground below the window for at least 5 s or past the end of the video recording) or died right after the collision. We used the absolute difference value of horizontal angle of approach relative to the perpendicular, ignoring left or right direction. We excluded vertical angle of approach as most birds flew approximately parallel to the ground and our camera set up did not allow us to quantify vertical angle accurately. Time of the day was transformed into decimals. None of our independent continuous factors showed any significant correlation between them (Pearson product moment correlation $r < 0.15$, $P > 0.40$). We focused the results on assessing whether the independent factors were significantly ($P < 0.05$) accounting for the variation in the dependent variables as well as presenting their effect size (and confidence intervals). Effect sizes were expressed in the odds ratio scale (*Sperandei, 2014*), which required that we centered our independent variables for the statistical analyses to standardize the effect sizes to the same scale.

**Table 1 Bird species represented in the collision and near-miss events.**

| Species name | | Event count |
|---|---|---|
| Blue Jay | *Cyanocitta cristata* | 13 |
| Northern Cardinal | *Cardinalis cardinalis* | 9 |
| Baltimore Oriole | *Icterus galbula* | 1 |
| Rose-Breasted Grosbeak | *Pheucticus ludovicianus* | 4 |
| Downy Woodpecker | *Dryobates pubescens* | 3 |
| White-Breasted Nuthatch | *Sitta carolinensis* | 1 |
| Common Grackle | *Quiscalus quiscula* | 4 |
| Black-capped Chickadee | *Poecile atricapillus* | 1 |
| Unknown | *Passeriformes sp.* | 2 |
| | Total | 38 |

However, figures were presented using the probability scale and the uncentered independent factors to facilitate interpretation. Statistical analyses were conducted using R 4.2.0 (*R Core Team, 2022*). We present descriptive results from our video analyses with means and SDs reported for each coded variable.

### Ethics statement

This study adhered to the guidelines of the University of Western Ontario and the Canadian Council on Animal Care (CCAC) and all animal handling procedures were approved by the institution's Animal Care Committee (protocol 2019-118).

### RESULTS

In total, we recorded 38 events comprising 29 collisions, in which a bird contacted the glass, and nine near-miss events, in which birds approached the glass but appeared to change flight trajectory to avoid it. We excluded two recorded collisions from the flight velocity analysis because it seemed likely that the birds were attacking their reflection on the glass and not attempting to fly through; one additional recording (#19) was excluded because it did not capture enough footage before impact to be useful for measuring changes in the position of the bird. At least nine other recorded events in which a bird flew close to the window, but did not contact the glass nor appeared to be on a collision course, were excluded from the analyses. The flight paths taken by each bird shown as collisions and near-miss events are provided in the Supplemental Material. Within collision events, four birds impacted screens covering a glass door (all flew away immediately), 13 birds collided with a glass door, and 12 birds collided with the upper transom windows above the glass doors. Interactions between birds of the same or different species prior to a collision or near-miss event were observed 13 times. Where footage allowed a clear view of the position of the bird's head in the moments leading up to impact, at least eight recordings showed birds flying with their beak pointed forward towards the glass. We did not find any evidence of birds turning their head during approach for near-miss or collision events. We provide a list of bird species that appeared in the recordings in Table 1.

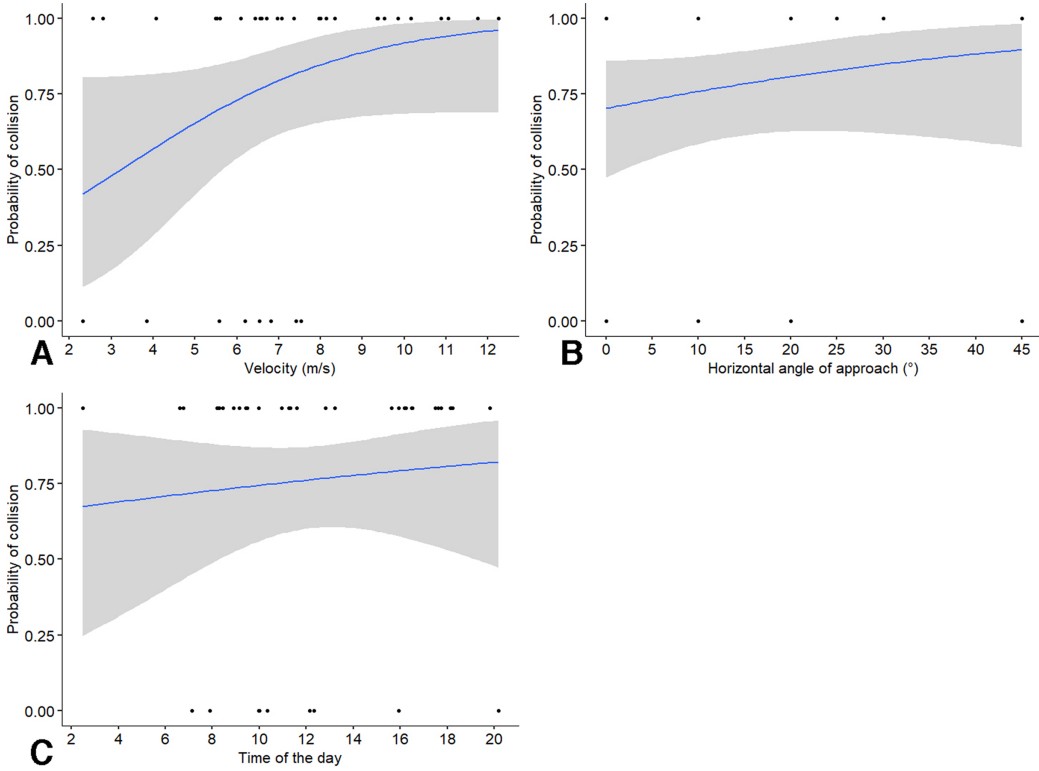

**Figure 2 Probability of birds colliding with windows in relation to independent factors.** Probability of birds colliding with windows in relation to: (A) approach velocity, (B) absolute horizontal angle relative to the perpendicular direction of approach, and (C) time of the day. Some data points may be overlapping and not visible in the figure.

Of the recorded collision events, two collisions resulted in an immediate fatality (one a Northern Cardinal, the other a Downy Woodpecker) as captured by the cameras and confirmed by the building occupants. Three birds were stunned and grounded upon impact but flew away within a few minutes. The cameras stopped recording within seconds following impact, but the homeowner confirmed observing when birds recuperated and flew away minutes later, out of the recording range for the videos. The other 24 collision events resulted in birds immediately flying out of the frame; the outcome for those birds could not be determined. For analyzed collision events ($n = 26$), mean flight velocity leading up to impact was 7.24 m/s, median = 7.02 m/s, SD = 2.45 m/s, range 2.31 to 12.25 m/s; mean absolute angle of approach was 16.53° from perpendicular to the window, median = 10.00°, SD = 19.45°, range 0° to 45°; and mean time of the day (decimal) was 12.39 h, median = 11.46 h, SD = 4.32 h, range 2.47 to 20.18 h.

We first assessed the probability of collision with a window when birds were on a collision course. We found that the probability of collision was not significantly affected by velocity ($\chi^2_1 = 1.52$, $P = 0.218$; odds ratio = 5.37, CI [0.31–92]), angle of approach ($\chi^2_1 = 2.06$, $P = 0.152$; odds ratio = 2.76, CI [0.57–13.40]), or by time of the day ($\chi^2_1 = 1.40$, $P = 0.236$; odds ratio = 0.19, CI [0–5.37]). Of the three variables, velocity of the approach had the highest effect size, with an increase in 1 m/s speed leading to a 5.37 increase in the

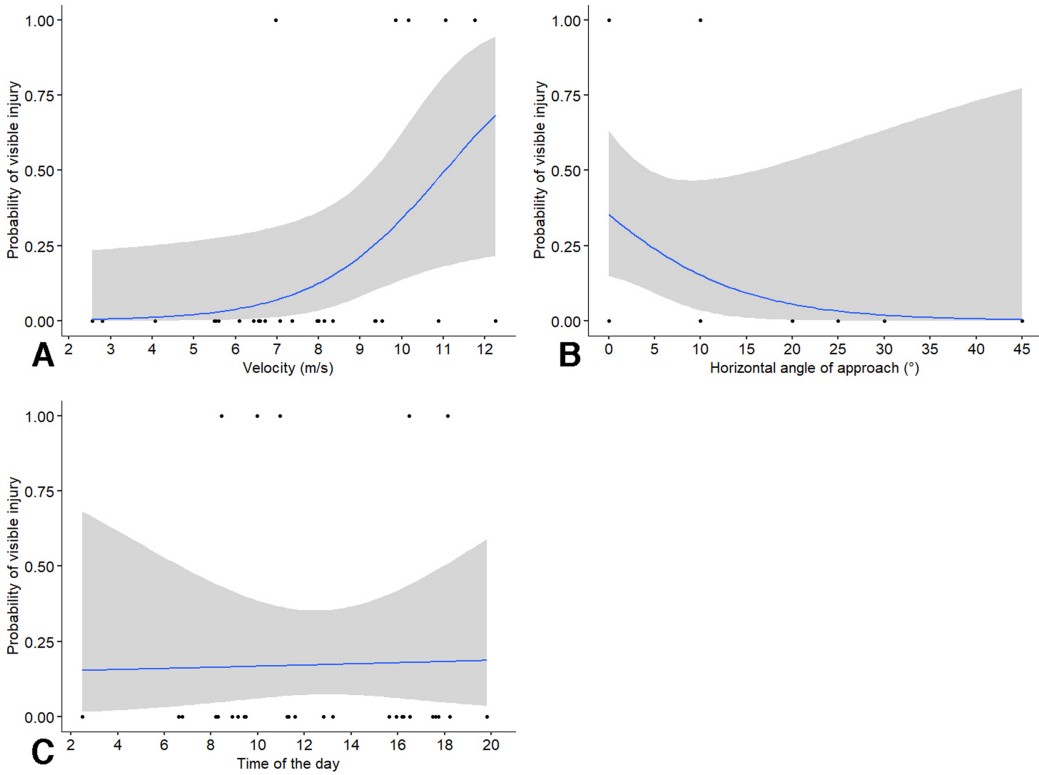

**Figure 3 Probability of birds enduring a visible injury (stunned, death) after colliding with windows in relation to independent factors.** Probability of birds enduring a visible injury (stunned, death) after colliding with windows in relation to: (A) approach velocity, (B) absolute horizontal angle relative to the perpendicular direction of approach, and (C) time of the day. Some data points may be overlapping and not visible in the figure.

odds of a collision. However, the effects of velocity were non-linear as shown by probability of collision (Fig. 2A), with the highest increase in the probability of a bird colliding with the window relative to speeds in the range of 2.5 to 8.5 m/s from our visual examination of the output. The probability of collision increased with angles greater than the perpendicular, but the changes were not very pronounced (Fig. 2B). Finally, the changes in probability of collision with time of the day appeared very minor (Fig. 2C).

After birds collided with a window, we assessed the probability of collision of a visible injury (*i.e.*, bird being stunned for a few moments before resuming flight or dying). We found that the probability of a collision leading to a visible injury was significantly affected by velocity ($\chi^2_1$ = 8.78, $P$ = 0.003; odds ratio = 12.8, CI [0.86–192]) and angle of approach ($\chi^2_1$ = 8.70, $P$ = 0.003; odds ratio = 0.03, CI [0–1.63]) but not by time of the day ($\chi^2_1$ = 2.32, $P$ = 0.128; odds ratio = 3.02, CI [0.56–16.3]). The statistical results should be interpreted with care because the model yielded a singular solution. Again, velocity of the approach had the highest effect size, with an increase in 1 m/s speed leading to a 12.8 increase in the odds of a visible injury. The probability of visible injury showed it highest rate of increase at speeds above approximately 7 m/s from our visual assessment of the output (Fig. 3A). The probability of visible injury showed the highest rate of decrease with

angles of approach from 0° to 20° (Fig. 3B). The change in the probability of collision did not vary much with time of the day (Fig. 3C).

## DISCUSSION

We documented 29 collisions and nine near-misses. Our data reveal previously undocumented variation in birds' pre-collision behaviour. We predicted that birds flying at higher speed and approaching the windows at approximately perpendicular angles would be more likely to suffer a visible injury or be killed immediately by the impact. We found support for this prediction, as flight velocity and angle of approach were both related to the outcomes following collisions.

### Flight characteristics and lethality of collisions

Velocity of approach yielded the highest effect size of the independent variables studied. The probabilities of collision increased with velocity faster in the range of 2 to 9 m/s and then leveled off. Additionally, birds flying faster than 7 m/s faced a higher probability of suffering a visible injury. However, our study involved a small sample size, especially for video recordings of birds that were stunned or killed by a collision and for birds that nearly avoided making impact, so the importance of the observed difference in flight velocity should be interpreted with caution. Our results support the idea that the flight velocity and angle at which a bird makes impact with a window may influence the odds of a collision as well as the outcome of a collision (injury, death, *etc.*).

Birds admitted to rehabilitation centres following a window collision often have patterns of soft tissue injury sustained by the head and body as force is absorbed upon impact (*e.g.*, *Hudecki & Finegan, 2018*). Yet, there has been no research to date aimed at characterizing the mechanics of collisions and understanding how these injuries occur. It seems likely that the severity of injury is related to the amount of force that the bird absorbs when it collides. Modelling collision dynamics in three-dimensional space requires more precise measurements than we were able to obtain with our video cameras. However, based on our empirical data, we can conceptualize how energy is distributed when birds impact a window under different configurations. When a bird strikes a window, some of the kinetic energy generated by the impact will be absorbed by the window, while the rest is directed outward at the bird. Generally, if a soft object (*e.g.*, a bird) strikes a harder object (*e.g.*, a window), most of the energy available for producing post-collision (final) velocity is stored in the soft object. Part of that energy pushes the bird outward from the window (*i.e.*, normal force), while the rest of the force is absorbed by the bird's body, compressing its tissues, and potentially causing injuries such as ruptures of blood vessels and broken bones. In theory, the ratio between these two components of force acting on the bird post-collision may vary with the duration of the impact, resulting from the extent that the bird's body compresses. The angle at which a bird approaches the window may also affect the amount and direction of force its body experiences upon impact. A collision at a perpendicular angle to the window (*i.e.*, a head-on collision) will result in momentum being conserved along only one axis. In other words, the momentum of the bird as it flies forward into the window will be transferred and redirected outward at the bird. However, a

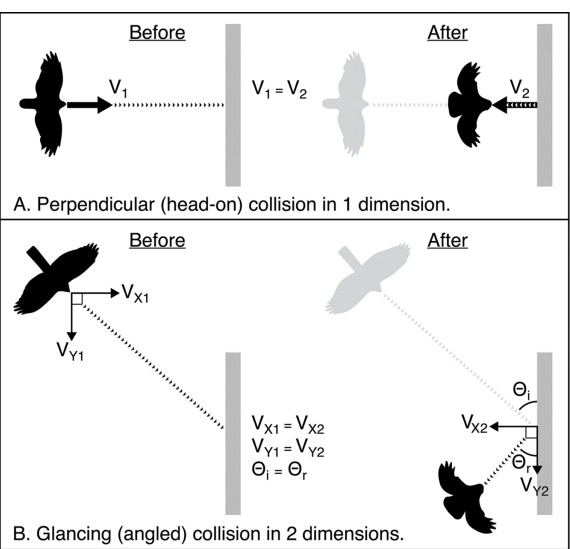

**Figure 4** **Schematic showing how momentum is conserved in two elastic collision scenarios from an overhead view.** Schematic showing how momentum is conserved in two elastic collision scenarios from an overhead view. Here we compare initial velocity (V1) and final (V2) velocity. The window is shown in grey. Dotted lines represent the path of the bird. Arrows represent vectors of the components of initial and final momentum. (A) A perfectly perpendicular bird-window collision where momentum is conserved in one dimension. (B) A glancing collision where momentum is conserved independently in two dimensions. In elastic collisions, the angle of incidence $\Theta_i$ is equal to the angle of reflection $\Theta_r$.

collision at an oblique angle, either in the horizontal or vertical plane, would result in components of momentum being conserved along multiple axes independently (Fig. 4). A bird that collides at a glancing angle may experience less force, reducing the likelihood of it sustaining injury. For example, all five of the collisions we recorded that resulted in birds being stunned or immediately killed involved angles of approach that were approximately perpendicular to the glass, while collisions that resulted in birds immediately flying away included more variable angles of approach.

More energy absorption by the bird's body may result in lower final velocity and displacement, but this could produce more damage to the bird's tissues. Perhaps variation in morphology and body composition, and resulting differences in how much energy is absorbed by soft tissues during collisions, could explain why collisions are apparently more lethal for some birds than others (*e.g.*, *Nichols et al., 2018*). In the real world, collisions occurring in three-dimensional space may be complicated by other factors, such as the bird's behaviour and wind acting on the bird. Future work using cameras with higher frame rate and resolution could be used to observe differences in the durations of birds impacting windows. One way that transfer of energy during bird-window collisions could be studied further is by simulating crash tests using objects with similar physical and mechanical properties to the body of a bird.

## Environmental variables and collision risk

Buildings with larger window surface area generally pose a higher risk of collisions for birds (*Hager et al., 2013*; *Elmore et al., 2021*). However, in our study we found that nearly half of the collisions we observed were with the smaller transom windows above larger glass doors. This suggests that small windows can also pose a significant risk of bird-window collisions. It seems likely that other factors may be important for explaining birds' behaviour in this setting. For instance, when birds take off from the feeders or surrounding trees, they may aim their flight towards the smaller transom windows because they perceive it as safer cover than the larger sliding door windows below. Alternatively, birds may simply prefer to fly higher and are therefore more likely to hit upper windows.

Although the configuration of our cameras did not allow us to continuously monitor birds' behaviour in the space surrounding the study area, the recorded footage revealed conspecific and interspecific interactions among birds directly preceding a collision in at least 12 instances. These included birds chasing each other, competing for access to feeders, and escaping predation. Our observations support the idea that distractions from social interactions and interactions with predators, that occur more frequently near bird attractants, may influence birds' behaviour and ability to detect and avoid collisions with windows during fleeing or chasing.

## Considerations for future research

Recording bird-window interactions passively in the field allows for behavioural observations that might otherwise require prohibitively invasive methodology with captive animals. In the present study, we captured nine near-miss events in which birds approached but did not collide with the windows. The total number of near-miss events was probably higher, but some events did not trigger video recording because the camera lenses and sensors were not oriented to observe birds several metres from the glass where detection may first occur. Future studies aiming to record near-misses more effectively could deploy more cameras oriented outwards, or combine cameras with other positional sensors. Given that most of the collision and near-miss events that produced triggers and were successfully recorded involved medium to large passerines (*i.e.*, with body mass exceeding 40 g), it is possible that events involving smaller birds may be less likely to be picked up by the cameras. To limit detection bias, future studies using cameras for monitoring bird collisions should conduct preliminary trials to calibrate the camera trigger sensitivity.

The perceptual mechanisms involved in successful detection and avoidance of windows by birds in flight are not clearly defined (see *Martin, 2011* for a discussion of bird vision and collisions with obstacles in general). Characterizing perceptual processes that birds use to avoid collisions could have implications for designing effective window collision prevention methods. These perceptual processes may interact with various factors, such as the morphology of the bird, interactions with a predator, or the rate of visual information being influenced by flight speed (*Martin, 2017*; *Jackson et al., 2020*, *Bhagavatula et al., 2011*). Recording pre-collision behaviour could provide an empirical basis to develop understanding of how birds see windows and how window collision deterrent

technologies, such as window markings, could function. For example, birds may rely on specific parts of their visual field and depend on visual information from their surroundings prior to takeoff or to detect and avoid obstacles mid-flight. Visual sensitivity that is necessary for birds to detect an obstacle may vary across parts of the visual field and depending on the distance from the object. Birds with laterally-positioned eyes, such as many passerine species, may turn their heads to orient the high-acuity fovea at the centre of one eye towards stimuli of interest. No research to date has explored whether the conspicuousness (or probability of detection) of window collision deterrents to a bird could be affected by the materials being viewed by the bird through different parts of the visual field prior to takeoff or during flight. Our camera system did not provide the quality of footage necessary for analyzing birds' head or gaze orientation upon approaching the glass in detail, but in all of the clips where head orientation was evident, we found no signs of birds turning their head upon approach. Future research simulating bird flights and collisions in controlled settings may yield more data to indicate how birds orient their eyes to detect objects and avoid collisions in flight.

The risk of bird-window collisions is elevated in environments with bird attractants such as feeders, bathing stations and native plants (*Klem et al., 2004*, *Bayne, Scobie & Rawson-Clark, 2012*, *Kummer & Bayne, 2015*, *Kummer, Bayne & Machtans, 2016*). Most bird-window collisions that we recorded using cameras at a residence with bird attractants left behind no trace and were not detected by the building occupants who were present for most of the study period. Our results are consistent with *Kummer & Bayne (2015)* who reported similar collision monitoring research carried at multiple residences with and without bird feeders. Their experimental protocol involved homeowners performing daily perimeter checks, and accepted forms of evidence of collisions beyond injured birds and carcasses, including homeowner observations through hearing or seeing a collision, as well as smudges, feathers or blood left on the glass. They found that the total number of bird collisions that occurred at several residences was comparable to our study, and that only 7.5% of all the collisions that were reported resulted in an observed fatality, whereas 92% of birds that collided were observed to fly away immediately or their status following the collision was unknown. This provides support for the notion that bird-window collisions could be far more common in residential settings than is realized by homeowners. If a large proportion of collision events are not observed, this could have implications for the reliability of datasets built using collision-reporting community science tools (*e.g.*, *Kummer, Bayne & Machtans, 2016*).

## CONCLUSIONS

Our study provides confirmatory empirical evidence that bird-window collisions occur frequently at residences with untreated glass windows near bird attractants. Yet, most collisions that we recorded were not observed by the building occupants and birds flew away leaving no trace. At least some of these birds may suffer morbidity or mortality far from the window site. Additional studies that can characterize how flight characteristics affect the force experienced by a bird during a collision will provide better estimates of the negative effects on birds that are able to fly away following collision. We conclude that

building surveys and other monitoring methods that rely solely on carcass detection vastly underestimate the total number of window collisions by birds (*Loss et al., 2014*).

Given our finding that birds' flight velocity prior to impact predicts the lethality of collisions with windows, we support recommendations that bird attractants should be placed close to windows to reduce the available space where birds can gain speed (*Klem et al., 2004*). Furthermore, although this study did not examine the effectiveness of bird collision deterrents, the risk of collisions at existing and new structures can be reduced through the application of "bird-friendly" materials such as fritted glass, window film or other visual markers (*e.g.*, *De Groot et al., 2022*). In the summer of 2021 following the conclusion of this study, the homeowner retrofitted the windows in the study area using a patterned commercial film consisting of a grid of dots spaced 5 cm apart. Based on their own monitoring in the time since the retrofit, the homeowner reported that they have not observed any additional bird collisions with the treated windows.

Future monitoring studies using video recording cameras in other residential settings, as well as mid and high-rise buildings, could yield new information about dynamics of bird-window collisions under different scenarios or involving different species. Our successful deployment of a basic home security video recording system to record bird-window interactions demonstrates the feasibility of scaling up bird collision monitoring efforts using camera technology, such as through community science. Many buildings already use similar camera equipment for security that could be adapted to monitor windows for bird collisions, especially while risk is elevated during migration periods or as birds visit nearby attractants.

## ACKNOWLEDGEMENTS

The authors thank Sherri Fenton for her support with carrying out the research.

### Funding

This study was funded by a Natural Sciences and Engineering Research Council of Canada Discovery Grant to Scott A MacDougall-Shackleton (RGPIN-2018-05658). The funders had no role in study design, data collection and analysis, decision to publish, or preparation of the manuscript.

### Grant Disclosures

The following grant information was disclosed by the authors:
Natural Sciences and Engineering Research Council of Canada Discovery: RGPIN-2018-05658.

### Competing Interests

The authors declare that they have no competing interests.

## PeerJ

## Author Contributions

- Brendon Samuels conceived and designed the experiments, performed the experiments, analyzed the data, prepared figures and/or tables, authored or reviewed drafts of the article, and approved the final draft.
- Brock Fenton performed the experiments, authored or reviewed drafts of the article, and approved the final draft.
- Esteban Fernández-Juricic analyzed the data, prepared figures and/or tables, authored or reviewed drafts of the article, and approved the final draft.
- Scott A. MacDougall-Shackleton conceived and designed the experiments, analyzed the data, authored or reviewed drafts of the article, and approved the final draft.

## Animal Ethics

The following information was supplied relating to ethical approvals (*i.e.*, approving body and any reference numbers):

This study adhered to the guidelines of the University of Western Ontario and the Canadian Council on Animal Care (CCAC) and all animal handling procedures were approved by the institution's Animal Care Committee (protocol 2019-118).

## Data Availability

The code used for our statistical analyses conducted in R Studio, and the raw data from coding of the videos that was included in the analyses, are available in the Supplemental Files.

## Supplemental Information

Supplemental information for this article can be found online at http://dx.doi.org/10.7717/peerj.14604#supplemental-information.

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
