# Peer review of "Opening the black box of bird-window collisions: passive video recordings in a residential backyard"

_PeerJ, doi:10.7717/peerj.14604_

## Round 0.1 · original submission · Major Revisions

Thank you for submitting your interesting article to PeerJ. Two reviewers both recommend major revisions.

Reviewer 1 focuses on the context of the study, specifically justifying the hypothesis, more tightly focusing the Discussion, and elaborating on the implications for reducing damaging collisions. I think that some of the justification is presently included in the Discussion and could be condensed and moved to clarify the hypotheses.

Reviewer 2 has made numerous helpful suggestions. These include comments that echo those of Reviewer 1. Below, I provide on my perspective on a few of the suggestions.
• Additional discussion of the role of social interactions or other distractions in collisions. I found your mention of this factor interesting and relevant to future research, but investigating this cause was not a specific goal of your study, and the set-up was not designed to quantify it. It is likely that limited attention means that animals engaged in fleeing one threat may be less able to detect an obstacle or a different threat. There is some literature on vigilance in relation to attention in fighting or territory defense (and I have anecdotally observed fish suddenly fleeing one threat to be caught by a nearby ambush predator), but I don't think you should elaborate beyond what you have already done.
• Camera technology. I agree with the reviewer that too much space is devoted to this topic. However, insights gained from your experience should not be lost. I suggest reducing this topic to a single paragraph devoted to the value and limitations of the technology you used and your experience with alternative technology that was less successful. Future needs and opportunities can be included in the Conclusions section (see below).
• I agree with the reviewer that Figs. 2 and 3 could be moved to supplemental material and that the four data figures could be a single, 4-panel figure. (Take note of PeerJ guidelines for panel identification.)

Editor's Comments
You can consider these as a third review, i.e., make changes where they are appropriate and provide specific explanations where they are not.
L31, 31. Providing actual numbers here would increase the impact of the Abstract.
L404. For some readers, blue jays and cardinals may not be what they imagine by 'larger' birds. A small clarification is needed.
L377. The observation that most bird collisions were not detected by occupants of the home and would not have been detected by carcass inspection, but might still involve injuries seems important for future studies. Consider including this point in the Abstract. At present, the Abstract devoted almost nothing to your interpretation of your findings.
L436. PeerJ does not want a 'second abstract' in the Conclusions. Rather, use this section for unresolved questions/gaps/future directions. The present Discussion has a number of such items which could be moved down to shorten the Discussion and replace the current repetition.
References. Congratulations on one of the most error-free Reference sections I have seen in a long time! The only formatting errors I noticed were a couple of journal titles not consistently in italics and one period (where elsewhere you used commas).
Minor grammar and wording. I have attached a pdf with highlights to indicate a number of minor problems.

Reviewer 1 ·

Basic reporting

No comment.

The ms. needs a considerable overhaul before considering the specific point of reporting, design, and validity. See "additional comments".

Experimental design

no comment

Validity of the findings

no comment

Additional comments

Sorry to say I do not think the ms. is ready for publication, and in this from I do not think it is worth going into detailed discussion of design, validity, or basic reporting.

There are a number of typos (repeated words and phrases) that need attention. They should have been detected by the authors.

My main concern is that the authors state that they are testing an hypothesis /prediction that, “the risk of being killed or incapacitated (i.e., stunned or injured) by a collision with a window, is related to the bird’s flight velocity and angle of approach to the window. We predicted that higher flight speed, and angles of approach approximately perpendicular to the window, are more likely to result in injurious or lethal collisions”. This sounds straight forward but it is not based upon an a priori argument, no evidence is given why this might be the case. There should be an argument made in the introduction as to why flight velocity and angle of flight relative to the glass surface should influence the outcome of a collision. This needs to be juxtaposed with other factors, for example, mass of the bird, light level, angle of the sun, density of background vegetation, etc. that all could conceivably influence collision outcome. For the authors to state their current prediction there needs to be information given on all the factors that could contribute to the outcome of a collision, then they should argue why they suggest that flight speed and angle of approach might be key contributors to outcomes.

The final sentence of the introduction says, “We discuss the implications of characterizing pre-collision behaviour for designing effective collision prevention methods”. I have not found any such discussion and it is not in the final Conclusions. The authors suggest, “The risk of bird-window collisions at existing and new structures can be reduced through the application of bird-friendly materials such as fritted glass, window film or appropriately spaced visual markers”. However, this is not a conclusion drawn from their data, rather it is based upon an anecdotal post study observation. So, do the authors have anything to say about effective collision prevention methods based upon the results that they present?

The Discussion is far too long, it rambles around a number of topics and there are no subheadings to collect ideas and to signal to the readers where the narrative is heading.

Finally, the title is not helpful. What does “opening the black box” mean? It is a trivial eye-catcher, not a serious attempt to communicate what the work is about.

My overall conclusion is that the authors need to review the context in which they justify their study, make some clear predictions based upon previous findings, and narrow down the discussion to a small number of points (presented using sub-headings) and finally discuss the implications of their actual results for the design of effective collision prevention methods.

Reviewer 2 ·

Basic reporting

The authors have presented their research in clear, well-written English and have followed professional standard of courtesy and expression. There are a few instances in the text where statements are presented with ambiguity (see comments for L86-87, L189, L193-194, L446-447)

This manuscript contained a well-written introduction that briefly highlighted the relevant background information needed to understand the objectives and relevance of the research. Associated with the earlier comment on ambiguity, there are sentences in the introduction and discussion that are lacking support from prior research (see comments for L53-56, L60-62, L342-345, L429-430)

The manuscript was presented in a professional structure. All figures are relevant to the content of the article, are of sufficient resolution, and are described and labeled. Raw data and code associated with the results have been made available to reviewers. There are minor revisions required to the code that can improve reproducibility of the results:
- Typo in line 22 of script: the variable “horizangleabs” is not in the data. I believe the authors are using the variable “horizangle” in this case.
- Include the libraries used in these analyses are the beginning of the script. I have been able to reproduce the results using libraries “afex”, “psych”, “ggplot2” and “optimx”.
- Include a metadata file with descriptions of each variable in the dataset.

This submission is self-contained and represents an appropriate unit of publication, where the results presented are relevant to the hypothesis. While the discussion elaborates on the presented findings and is relevant to the research questions, there are elements of the discussion which are not clearly related to the results (see comments on discussion in the ‘Validity of findings’ section of this review).

Experimental design

The main objective of this research was to record the pre-collision behaviors of birds and determine their association with rates of window collision and bodily injury. The authors have presented a current knowledge gap in the literature (the lack of insight into the behaviours of birds that occur prior to a bird-window collision event) that highlights the relevance of their research in addressing this gap.

The authors have recorded behaviours data of birds for two years at one location. It is unclear why only one location was used, but given the global circumstances of these past 2.5 years and the authors’ explanations of their data limitations in the discussion, I cannot fault the authors for this.

I would suggest that the methods would benefit from further elaboration on their experimental design. Specifically, there are aspects of the camera setup that are not clear from the text and figures (e.g., the camera position relative to each window; relevant to both capture of bird behaviours and the calculation of the velocity and angle of approach variables).

Validity of the findings

The authors appropriately highlight the lack of information on pre-collision behaviours as a knowledge gap in the literature and design their study to address this gap. The data used to produce these results have been made available by the authors along with R code that analyzes these data. I comment on a few aspects of their calculations on velocity and angle of approach that require clarification (see comment to authors for L173-187 below).

In general, the authors provide discussion on the results and their relevance to the field of bird-window collision literature. However, there are both (i) important results that have not been discussed and (ii) discussion on points that draw away from main results.

Example for (i): Authors report that there were 15 instances of inter/intraspecific interactions prior to an observed event. This is a great opportunity to discuss the importance of animal aggression, competition, social behaviour, etc. on the risk of collision. However, this result was not discussed. (Lines 221-222)

Example for (ii): There are three paragraphs dedicated to discussing camera tracking technology (paragraph starting on Lines 391, 406, and 418). These points of discussion would be more appropriate to discuss had the study been designed to test the validity of tracking methods. In the presented study, these paragraphs take away from the main message of the effects of behaviours on collision and injury risk. These points would be better included as a brief mention of suggested improvements to the methods of this study.

I believe these issues can be resolved by careful construction of both the introduction and discussion sections of this manuscript, such that the hypotheses presented in the introduction are either supported or rejected by the results, and the accompanying discussion explains reasons to why that is the case.

Additional comments

I believe this research highlights a key aspect of the bird collision literature that has not yet been addressed. Additionally, I thank you for your well annotated code and accessible data – these are great for promoting both reproducibility of research and collaboration of science. Below, I have included my comments and suggestions on the manuscript. I look forward to reading the final version.

Introduction:
L53-56
This sentence states that numerous field studies have monitored bird-window collisions but is followed by no citations. Please add examples of such studies to support this claim.

L60-62
Need citations to support this statement of the empirical value of bird monitoring studies. Give examples of the types of studies you are referring to.

L86-87
Clarify the term “relative lethality”. I assume this refers to the ratio of birds that have collided with windows to birds that have experienced an injury from such collision.

L103
This is the first time we hear about velocity and approach angle. The relevance of these two variables on bird-window collision rates should be presented earlier in the introduction along with explanation on the hypotheses that led to your predictions (faster flight leads to more injuries, greater angles of approach lead to more injuries).

L105
Similar issue (as L103 comment) for “time of the day” variable. Additionally, is this the most appropriate metric for ambient light? The amount of ambient light within a given day varies across a year (e.g., in London ON, 8:00 on Feb 1 would usually have less sunlight than 8:00 on July 1). In the methods, the authors mention that data were collected across two years, where I suspect there would be an interaction between time of day and day of the year. Have you accounted for this in the analyses? If so, please add this explanation in the methods section.

Another note: Given the small sample size of this study, would it be possible to use the recoded videos to extract a value of ambient light (in lumens) present in the environment for each event? This could be an alternative, more accurate way to account for the variation in ambient light throughout the sampling period.

Materials and Methods:
L109
Could you include the coordinates of the site? I understand if this was excluded for concerns of privacy for the homeowner(s).

L111-112
What led you to presume the high abundance of birds? Was this due to previous research or local recorded of bird abundance? A brief elaboration of this presumption would support this statement.

L145-148
Can you report the sensitivity thresholds that were set for visual and audio triggers? This is important for replication purposes, even if locations differ. Another option would be to include a more detailed explanation of the protocol you followed for setting up appropriate thresholds given your sampling location, such that future research could follow/build upon the study’s protocol.

L157
What is the continuous video recording function? Is it important to state in this case (i.e., could be a proprietary software function given the choice of camera) or could this be replaced by simply stating that two camera continuously recorded behaviours?

L163-165
Seems that something unsavoury happened to this sentence. Likely a typo.

L173-187
As a reader, I would want an indication on the accuracy and precision of these calculations. Capturing 3D movement using cameras can be challenging, where a video in the x and y planes will not capture movement in the z plane. Did you compare your calculations across multiple cameras, or use multiple cameras to calculate less-biased variable values?

A quick look as the raw data shows that velocities are variable, but angles of approach seem to be grouped into a handful of values (specifically looking at angles of 45 (n=8) and 90 (n=17)).

Including an example of a recorded event, along with the velocity and approach angle calculations associated with such event would, would better explain the methods for generating both variables.

L182-184
Why was horizontal approach angle chosen instead of/in addition to vertical approach angle?

L189
The first sentence of this paragraph is extremely vague. Please provide information about which results are being referred to in this case and where they’re being presented.

L193-194
“Negative statistical implications” is too vague. Instead, mention explicitly what implications you’re referring to (e.g., that the random effect accounts for the non-independence of samples in the model). Additionally, replace “decided to include” with “include” – the inclusion of a random effect should be considered an appropriate statistical practice rather than a personal choice.

L196-197
I am not sure probability is the most appropriate term to use here and recommend a change in the variable name (such as dropping the “probability” prefix). The use of “probability” refers to a chance of an event occurring, with a range of outcomes between 0 and 1. Both response variables (occurrences of collision and injury) are binary variables of either 0 or 1. While 0 and 1 can fundamentally be considered probabilities, it is misleading to suggest that there are other probabilities in the data.

Results:
L210
The total number of events should be 38.

L210-212
Collisions and near misses should be first defined in the methods section.
Also, how was a “change in trajectory” quantified? This should be clarified as it determined which events were included in the dataset and which ones were excluded (mentioned later on L216-217). One option is to provide a video example of both cases in the supplementary material.

L216-217
Please provide a sample size for the total number of excluded events.

L221-222
This mention of interactions between birds prior to an event is relevant to the main question asking on how pre-collision behaviours influence collision/injury rates, but is not discussed in the discussion. Include an explanation in the discussion about the ways that inter/intraspecific interactions can influence the risk of window collision.

L222-224
Add the sample size for the number of video clips of bird colliding with windows with beak pointed forward.

L230
The mention of the homeowner’s account of bird survival should only be mentioned if the birds flew out of the recording range for the videos, otherwise the raw recorded data would be the most validated source. If the former is the case, mention that these three birds were grounded outside of the recording field of view.

L239-240
The importance of 7.5m/s is unclear other than being an arbitrarily chosen value that describes collision rates of the sample. Consider explaining the importance of this value, or removing the result for this section and the discussion.

Discussion:
L342-345
Research by Jackson et al found that beak length is related to collision rate, and may be explained by visual obstruction to the bird (https://www.biorxiv.org/content/10.1101/2020.07.20.212985v1).

L402-403
Were there preliminary studies that were performed to measure any biases in bird species detection from your camera setup? If so, briefly mention how sensitive the set up was. If not, recommend that future studies should conduct preliminary trails to determine whether detection biases are present and way to reduce such biases.

L408-410
Present examples of behaviours that would differ between tall buildings vs. residence buildings and reasons why you would expect such a difference.

L429-430
Are 29 collisions across two years of recording considered frequent? Cite comparable literature here to support this claim.

Conclusion
L446-447
The conclusion that building surveys and monitoring programs vastly underestimate the number of bird-window collisions requires more clear support within the manuscript. Include citations of previous research values and compare with your results.

Figures
I recommend incorporating the four ggplot figures into one figure with four panels, ordered figure 4a (top left), figure 4b (top right), figure 5a (bottom left), and figure 5b (bottom right).
Figure 2 and 3 seem more appropriate as supplementary figures, since their descriptive intent is referred to as supporting information, not a main result.

---

## Round 0.2 · Minor Revisions

Thank you for your thorough response to my comments and those of the reviewers. Reviewer 1 now considers the manuscript ready for publication. However, Reviewer 2 did a detailed review of the data and found a typo that may affect the results.

I have reviewed the manuscript and have a small number of minor corrections.

I really like this study and feel it makes a substantial contribution to the window-collision literature.

Minor corrections
L7. 29 + 9 = 38, not 37 (see also Table 1 total)
L155 indent paragraph
L256. Normally, scientific names should be provided at first mention. I think that they can be skipped here because they are in Table 1 that was just cited.
L262-265. Because of the high variability, it would be helpful to provide the range. Also, if the data are not normally distributed, medians and quartiles would be more appropriate than mean and SD.
L294. I think the sentence reads better if ‘making’ is deleted.
L333. I think ‘frame rate’ is two words.
L357. ‘prohibitively’ not ‘prohibitive’
L380. I am not certain of your precise meaning: alternatives are ‘. . . parts of the visual field, depending on . . .’ or ‘. . . parts of the visual field and depend on . . .’
L429 ‘reported that’
Fig. 1.
There could be confusion about where the windows are because of the different view. I suggest adding something like ‘Dark shaded area represents the house with a horizontal view of the transom windows and sliding doors inset on the right.’ There would be some redundant words to be removed from lines 2 and 3.
Also, is it intentional that the arrow associated with 122 in the lower right doors does not cover the entire space? It’s not clear if I missed something or you did.
Table 1 lists 37, not 38 events.

Reviewer 1 ·

Basic reporting

No comment

Experimental design

no comment

Validity of the findings

no comment

Additional comments

Thank you for revising and following general recommendations of the reviewers. What you have done and the reasoning behind your hypotheses are now clear.

Reviewer 2 ·

Basic reporting

no comment

Experimental design

no comment

Validity of the findings

no comment

Additional comments

GENERAL FEEDBACK
The authors have done a good job addressing my feedback on their manuscript. The inclusion of supporting literature in the introduction has improved their justification of their research and the structure of the discussion has significantly improved with the inclusion of the subheadings. However, there were still issues detected during this second round of revision that need to be addressed. Specifically, (i) there is a typo in the data that, once corrected, generated results different than those presented (see comment for DC2), and (ii) results which are not supported by analytical methods (see comment for L270-271). Because of these issues, I do not think this manuscript is ready for publication in this current state. Please take care in ensuring the quality of the data and results prior to considering any of the other comments included below.

DATA AND CODE
DC1. Correct the spelling of the optimx package.

DC2. Regarding the data: While the results are reproducible (from the provided data and code), I came across a typo in one species observation (lowercase “j” in one Blue Jay observation; row 34 in the .csv file). This erroneously generated an additional level in the species variable. Running the analyses after correcting this typo yielded a different output for the first GLMM:
CollNear GLMM:
- Velocity X^2 = 1.52, P = 0.217
- Angle X^2 = 1.96, P = 0.161
- Time of the day X^2 = 1.41, P = 0.236
- NOTE: There are now issues with the convergence of the model
Please correct this typo in the data, revisit the model, and update the results and discussion if changes occur.

DC3. Correct the sample size for cardinals in table 1 (there are 9 present in this dataset, not 8).


MAIN TEXT
L7-8. These numbers do not add up (29 + 9 = 38 events, not 3). Please correct.

L9-10. While you mention in your main text that some events were excluded, it seems confusing that these 25 events do not match to the 29 events mention before. Consider modifying the way this information is presented to reduce possible confusion.

L10-12. In this abstract, it is not clear why variation in reflection would drive variation in velocity and approach angles. Consider including a sentence near the beginning that explains/mentions your hypothesis.

L27-28. Suggestion: Remove these two sentences, as they are discussed later in the introduction (L54-99) and do not clearly guide the reader into the next paragraph.

L90. Momentum should be defined earlier in the paragraph (when it is first used).

L270-271. The response from the authors on a previous comment (first round of revision, comment on Results L239-240) did not address my initial concerns about the reported 7.5m/s value. I do not agree that this trend is “self-evident” (as suggested by the authors) but instead is a product of how the model output has been visualized in Fig 2A. For example, modifying the tick labels on the x-axis to all integers while retaining the scale of the axis changes where one interprets this threshold to lay, if at all (see modified code below). Please include a supporting analysis that quantifies this threshold, however you choose to define it (e.g., on the rate of change in the slope), or mention that you have visually identified this threshold without support of an analytical method.

>> Code to modify x-axis tick labels for Fig 2A (R version 4.2.0, modified from the script provided by the authors during peer-review):

ggplot(binarydata, aes(x = velocity, y = collnear)) + geom_point() +
stat_smooth(method="glm", method.args=list(family="binomial"), se = TRUE) +
labs(y = "Probability of collision", x = "Velocity (m/s)") + theme_pubr(16) +
scale_x_continuous(breaks=seq(2,12,2))

---

## Round 0.3 · accepted · Accept

I have checked the rebuttal and track changes text. I believe that the manuscript is now ready for publication. The authors have addressed the reviewer and editor's concerns and more. This will make an interesting and useful contribution.